# Impact of between-tissue differences on pan-cancer predictions of drug sensitivity

**John P. Lloyd**[1,2,3], **Matthew B. Soellner**[2,3‡], **Sofia D. Merajver**[2,3‡*], **Jun Z. Li**[1,3‡*]

**1** Department of Human Genetics, University of Michigan, Ann Arbor, Michigan, United States of America,
**2** Department of Internal Medicine, University of Michigan, Ann Arbor, Michigan, United States of America,
**3** Rogel Cancer Center, University of Michigan, Ann Arbor, Michigan, United States of America

‡ These authors are joint senior authors on this work.
* smerajve@med.umich.edu (SDM); junzli@med.umich.edu (JZL)

**Data Availability Statement:** All relevant data are within the manuscript and its Supporting Information files.

**Funding:** This work was supported by funds from the Michigan Institute for Clinical and Health

## Abstract

Increased availability of drug response and genomics data for many tumor cell lines has accelerated the development of pan-cancer prediction models of drug response. However, it is unclear how much between-tissue differences in drug response and molecular characteristics may contribute to pan-cancer predictions. Also unknown is whether the performance of pan-cancer models could vary by cancer type. Here, we built a series of pan-cancer models using two datasets containing 346 and 504 cell lines, each with MEK inhibitor (MEKi) response and mRNA expression, point mutation, and copy number variation data, and found that, while the tissue-level drug responses are accurately predicted (between-tissue $\rho$ = 0.88–0.98), only 5 of 10 cancer types showed successful within-tissue prediction performance (within-tissue $\rho$ = 0.11–0.64). Between-tissue differences make substantial contributions to the performance of pan-cancer MEKi response predictions, as exclusion of between-tissue signals leads to a decrease in Spearman's $\rho$ from a range of 0.43–0.62 to 0.30–0.51. In practice, joint analysis of multiple cancer types usually has a larger sample size, hence greater power, than for one cancer type; and we observe that higher accuracy of pan-cancer prediction of MEKi response is almost entirely due to the sample size advantage. Success of pan-cancer prediction reveals how drug response in different cancers may invoke shared regulatory mechanisms despite tissue-specific routes of oncogenesis, yet predictions in different cancer types require flexible incorporation of between-cancer and within-cancer signals. As most datasets in genome sciences contain multiple levels of heterogeneity, careful parsing of group characteristics and within-group, individual variation is essential when making robust inference.

## Author summary

One of the central goals for precision oncology is to tailor treatment of individual tumors by their molecular characteristics. While drug response predictions have traditionally been sought within each cancer type, it has long been hoped to develop more robust predictions by jointly considering diverse cancer types. While such pan-cancer approaches have improved in recent years, it remains unclear whether between-tissue differences are

Research – Postdoctoral Translational Scholar Program (michr.umich.edu; UL1TR002240 to JPL), Breast Cancer Research Foundation (brcf.org; to SDM), Michigan Institute for Data Science (midas.umich.edu; to JZL), and National Institutes of Health (nih.gov; NIH 1R21CA218498-01 to SDM and NIH R01GM118928-01 to JZL). The funders had no role in study design, data collection and analysis, decision to publish, or preparation of the manuscript.

**Competing interests:** The authors have declared that no competing interests exist.

contributing to the reported pan-cancer prediction performance. This concern stems from the observation that, when cancer types differ in both molecular features and drug response, strong predictive information can come mainly from differences among tissue types. Our study finds that both between- and within-cancer type signals provide substantial contributions to pan-cancer drug response prediction models, and about half of the cancer types examined are poorly predicted despite strong overall performance across all cancer types. We also find that pan-cancer prediction models perform similarly or better than cancer type-specific models, and in many cases the advantage of pan-cancer models is due to the larger number of samples available for pan-cancer analysis. Our results highlight tissue-of-origin as a key consideration for pan-cancer drug response prediction models, and recommend cancer type-specific considerations when translating pan-cancer prediction models for clinical use.

## Introduction

Tailoring cancer treatment to the molecular characteristics of individual tumors represents one of the central strategies of precision oncology [1–3]. Public repositories of drug response and genomics data in many cancers have facilitated the identification of somatic mutations that underlie variable drug response [3,4]; and drug response prediction analyses have expanded from analyzing known cancer genes to unbiased searches across the human genome [4,5]. More recently, prediction analyses have further incorporated data modalities beyond DNA, to include gene expression, epigenomic, and/or metabolomics data [4,6,7]. Drug response predictions using the tumors' molecular characteristics are frequently performed within a single cancer type, using primary tumors or auxiliary models derived from them, which are generally expected to share the primary tumors' vulnerabilities to specific anti-cancer agents–presumably as a result of tissue-specific oncogenic processes involving similar cell types. Meanwhile, drug response predictions can also be developed by considering multiple cancer types jointly, through *pan-cancer* analyses. The success of pan-cancer prediction models is predicated on diverse cancer types sharing a broad set of molecular vulnerabilities despite tissue-specific mechanisms of cancer initiation, progression, or drug-response.

Pharmacogenomic databases of patient-derived cancer cell lines now cover a broad range of cancer types and represent reusable pre-clinical models for finding the cell lines' innate characteristics that may contribute to their drug response profiles [8–15]. Using tumor cell line resources, many groups have employed DNA variants and gene expression levels to develop pan-cancer drug-response prediction models via a variety of computational methods, including regularized regression [8,16], random forests [17,18], neural networks [19], network modeling [20,21], quantitative structure-activity relationship analysis [22], and deep learning [23,24]. These analyses have offered many insights, including the importance of RNA expression and tissue context for pan-cancer predictions [17,25,26], higher accuracy of multi-gene classifiers (i.e. gene panels) compared to single-gene classifiers [18], and the suitability of cell lines as *in vitro* mimics of primary tumors [16,17].

Tissue-of-origin has a complex relationship with pan-cancer drug response predictions [14,15,25–27]. Some studies reported that pan-cancer models that include all available tissues may be outperformed by those that include only a well selected subset [25,26]. Others found that both drug response and molecular features (e.g. mRNA expression levels) often vary by tissue [27]. Thus, drug response can be predicted based on tissue type alone and tissue-specific molecular properties can drive the performance of a pan-cancer model without necessarily

being driven by inter-tumor differences within a cancer type [14]. It is also unclear if the prediction performance could vary among cancer types, a situation that would call for tissue-specific guidelines of applying prediction models. In clinical practice, while therapeutic decisions are often made solely based on cancer type, there is often the additional need, and potential benefit, to predict variable response among tumors within a cancer type.

We set out to examine the relative importance of between- and within-cancer type signals in pan-cancer drug response prediction models. Here, we analyzed data from two public cell line-based datasets [11,12], focusing on ten cancer types that were well-represented in both, and examined the performance of between-tissue and within-tissue models of pan-cancer drug response predictions for MEK inhibitors. In addition to cross-tissue effects, we evaluated cross-MEK inhibitor prediction models, and provided *in silico* replication by applying prediction models across datasets. Based on our results, we highlight key considerations for deploying pan-cancer drug response prediction frameworks and discuss the importance of jointly analyzing the contributions of both group (e.g. tissue type) and individual identity when interpreting the performance of prediction models.

## Results

### MEK inhibitor sensitivity across cancer types

To evaluate pan-cancer drug response predictions in pre-clinical tumor models, we utilized publicly available datasets of tumor cell lines described in a prior publication (referred to as Klijn 2015) [11] and the Cancer Cell Line Encyclopedia database (CCLE) [12]. Klijn 2015 and CCLE include 349 and 503 tumor cell lines, respectively, that have drug response data and RNA and DNA characterization, with 154 cell lines in common between the two datasets (**Fig 1A**), representing 10 cancer types defined by organ site (**Fig 1B**). Among the 5 and 24 drugs screened in Klijn 2015 and CCLE datasets, respectively, MEK inhibition was the sole target mechanism in common, with one MEK inhibitor (MEKi) screened in both datasets (PD-0325901; referred to as PD-901) and an additional MEKi unique to each dataset (Klijn 2015: GDC-0973; CCLE: Selumetinib). We therefore focused on predicting MEKi response, as prediction models could be evaluated for consistency across these two independent datasets. Moreover, MEK inhibition has shown promise for pan-cancer drug response predictions [17].

The molecular features of tumor cell lines available from the Klijn 2015 and CCLE datasets included treatment-naïve RNA expression levels (data from RNA-seq), single-nucleotide polymorphisms (SNPs; Klijn 2015: SNP microarray; CCLE: exome-seq), and copy number variation status for genes (CNVs; SNP microarray). RNA expression data were filtered to include only the 29,279 genes in common between the two datasets. We mapped SNP features to annotated genes, and determined if a gene contained a missense or nonsense SNP (binary variable for the gene: mutated vs. not mutated), while CNV status indicated if a gene was duplicated or deleted (3-level categorical variable: amplified, deleted, or no CNV). SNPs and CNVs had been filtered to exclude known germline variants based on the Exome Aggregation Consortium database [28], but likely still contained rare germline mutations alongside somatic mutations. In all, our DNA data contains mutation status (mutated / not mutated) for 12,399 genes and CNV-carrier status (amplified / deleted / no CNV) for 4,578 genes.

First, we compared MEKi responses of individual cell lines across cancer types and found clear differences ($\chi^2$ test of independence of sensitive vs. resistant cell line proportions across 10 tissues, all $p < 2 \times 10^{-9}$; **Fig 1C**). Skin, pancreatic, and colorectal cell lines were generally sensitive to MEK inhibition; while lymphoid, brain, ovary, and breast cell lines were generally resistant (**Fig 1C**). Still, other cancer types, such as the lung, stomach, and liver cancer lines, contain more mixed responses to MEKi (**Fig 1C**). For cancer types with mixed response, i.e. a

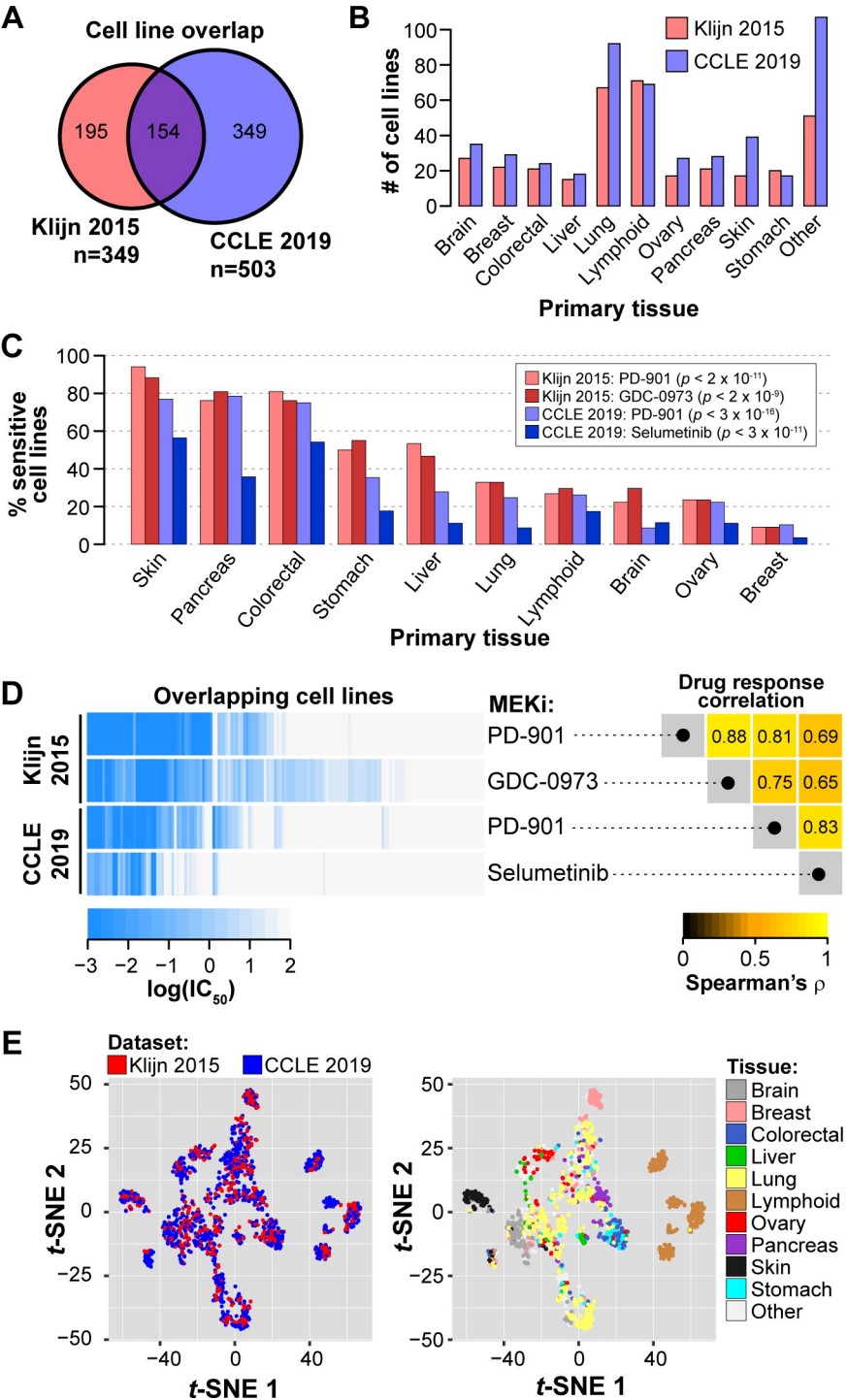

**Fig 1. Overview of between-cancer type differences of MEK inhibitor response and expression patterns in the Klijn 2015 and Cancer Cell Line Encyclopedia datasets.** **(A)** Overlap in cell lines with drug response and RNA expression and DNA variant data. **(B)** Counts of tumor cell lines with MEKi response data for 10 shared cancer types (n ≥15 in both datasets). **(C)** Proportion of MEKi-sensitive cell lines stratified by tissue. Cell lines were considered sensitive based on a threshold of $IC_{50} \leq 1$ nM. *P*-values from $\chi^2$ test of independence of sensitive vs. resistant cell line proportions across the 10 tissues. **(D)** Variability in the response to the MEK small molecule inhibitors. Left: Heatmap of MEKi response in $\log(IC_{50})$ for four data series (in rows). Each column corresponds to a single cell line with drug response data for both MEKi in both datasets; columns were hierarchically clustered. Right: Rank correlation of $\log(IC_{50})$ among the four data series in the left panel. **(E)** Scatterplots of the first two dimensions from *t*-distributed

stochastic neighbor embedding (t-SNE) analysis on transcriptome data. Each point represents a cell line and is colored by dataset (left) and tissue-of-origin (right).

wide range of observed responses within a cancer type, it would be especially challenging to make treatment decisions based on cancer type alone, hence there is enormous value and opportunity to develop individualized predictions. We further compared the response among the four data series: for two inhibitors in each of two data sources. For the 154 cell lines in common, MEKi response was highly correlated across distinct MEKi and between datasets (Spearman's $\rho$ = 0.65 to 0.88; **Fig 1D**). Within-dataset correlation of MEKi response for two different drugs (**Fig 1D**, right panel) was higher than cross-dataset correlation of the same MEKi (PD-901), a result that could be due to (1) continued evolution of cell lines, producing genuine biological differences for a cell line maintained in different laboratories, or (2) study-specific technical differences in measuring $IC_{50}$, through differences in cell viability assays and dose and duration of drug exposure. The observation that cell lines respond similarly to different MEK inhibitors supports the feasibility of cross-MEKi predictions.

We also compared gene expression profiles between cancer types in each of the two datasets by using *t*-distributed stochastic neighbor embedding (t-SNE) and principal component analysis (PCA) of standardized RNA expression levels (see Methods). Cell lines from Klijn 2015 and CCLE were jointly analyzed, and they occupy overlapping space in the t-SNE1/t-SNE2 and PC1/PC2 plots (**Fig 1E; Fig A** in S1 Text). The concordance between datasets supports our approach to develop and test prediction models across datasets (see below). Cell lines from the same primary tissue tend to be present in similar regions of the t-SNE and PCA space. Tissue-tissue correlations of RNA expression levels show that some tissues are distinct while others are similar to each other, and the patterns are reproducible across datasets (**Fig A** in S1 Text). These results indicate that cell lines derived from the same primary tissue have more similar transcriptomic profiles than cell lines from different tissues, a pattern that is consistent with previous cancer subtype analyses [29,30]. Given the effects of primary tissue on both MEKi response (**Fig 1C**) and transcriptomic profiles (**Fig 1E; Fig A** in S1 Text), it is plausible that between-tissue differences will be a major contributing factor in pan-cancer drug response predictions that consider RNA data, even in cases where tissue labels are not included in a prediction model. The Klijn 2015 and CCLE datasets provide the needed resources for us to investigate the relative importance of between-tissue and within-tissue signals in pan-cancer prediction models.

## Pan-cancer machine learning predictions of MEK inhibitor sensitivity

To build prediction models, we examined two ways of using the drug response variables: either taking $\log(IC_{50})$ as a continuous variable or dichotomizing drug response to a binary variable by categorizing cell lines as sensitive ($IC_{50} \leq 1\mu M$) or resistant ($IC_{50} > 1$ $\mu M$; see Methods; **Fig B** in S1 Text). For the former (continuous response variable), we evaluated two prediction algorithms: regularized regression and random forest regression. And for the latter (dichotomized response variable), we adopted two algorithms: logistic regression and binary random forest. As there are two MEKi in two datasets, for each of the four algorithms we developed four prediction models, named as $f_{K1}$ and $f_{K2}$ for the two inhibitors in Klijn, and likewise, $f_{C1}$ and $f_{C2}$ for the two inhibitors in CCLE. Only the cell lines unique to each dataset were used to train the models (n = 195 in Klijn 2015; n = 349 in CCLE), while the 154 cell lines in common were saved for validation (**Fig 1A**).

To assess performance, we examined two types of validation: 1) within-dataset validation: using the prediction model developed in a dataset (such as $f_{K1}$) to obtain predicted responses for the 154 common cell lines *in the same dataset*, and compare to their observed responses; and 2) between-dataset validation, using the model developed in one dataset to predict the

responses for all cell lines—both unique and common lines—*in the other dataset*, and compare to their observed responses. Model performance was evaluated with two metrics: rank correlation (Spearman's ρ) and concordance index between observed and predicted log($IC_{50}$) values. **Fig 2A** shows a workflow where a prediction model, named "$f_{K1}$", is trained on the data for PD-901 ($y_{K1}$) in the 195 lines unique to Klijn 2015 dataset (black arrows), and applied to predict (1) within dataset: $\hat{y}_{K(K1)}$ for the 154 common lines, using features in Klijn 2015, and (2) across dataset: $\hat{y}_{C(K1)}$ for all 503 cell lines in CCLE, including 349 unique and 154 common lines (dashed green arrows). Model performance is assessed by comparing the resulting predicted values from the $f_{K1}$ model to observed MEKi response (Klijn 2015: $y_{K1}$ and $y_{K2}$; CCLE: $y_{C1}$ and $y_{C2}$; blue dotted arrows). In total, four sets of models {$f_{K1}, f_{K2}, f_{C1}, f_{C2}$} were developed using four algorithms each (regularized, logistic, and regression and classification random forest), with predicted MEKi response compared to four observed MEKi response series {$y_{K1}, y_{K2}, y_{C1}, y_{C2}$}, with performance measured by two performance metrics (Spearman's ρ and concordance index) for 128 pan-cancer performance measures (**Fig 2B**). As two examples, **Fig 2C** showed the observed and predicted log($IC_{50}$) values for a within-dataset (regularized algorithm) prediction and a cross-dataset (logistic) prediction from the $f_{K1}$ model.

The 128 prediction performances are shown in **Fig 2D**, with the 4 algorithms arranged in different rows. For regularized and logistic regression algorithms, Spearman's ρ values ranged from 0.43 to 0.65, and concordance index values from 0.67 to 0.80, with similar performance between the two algorithms across different training and testing data (Mann Whitney U test, all $p \geq 0.86$). Random forest (RF)-based models (both regression and binary) exhibited Spearman's ρ performance similar to the regularized and logistic regression models, except for the $f_{C2}$ models: those trained on Selumetinib response and RNA and DNA features in CCLE, which exhibited lower performance relative to other algorithm × training set combinations (gray boxes in **Fig 2D**). In the following sections, we focus on predictions from the regularized regression and logistic regression algorithms, as these algorithms are simpler and performed similarly or better than the more complex random forest algorithms. Additional performance metrics for logistic and regularized regression algorithms are provided in **Table A** in S1 Text, including $r^2$ (range: 0.18–0.5), area under the curve–receiver operating characteristic (AUC-ROC; 0.74–0.86), normalized mean absolute error (11.8–28.7), and normalized root mean squared error (17.1–33.9).

We next compared the performance from the perspective of cross-dataset and cross-MEKi predictions. Prediction models applied across datasets (red and pink symbols in **Fig 2D**) performed worse (Spearman's ρ mean: 0.54; range: 0.46–0.62) than those applied within datasets (ρ mean: 0.59; range: 0.47–0.65, blue and cyan symbols) for regularized and logistic regression algorithms(U test, $p < 0.04$). When comparing within- and cross-MEKi predictions, we found marginally significant better performance for predictions of the same MEKi (ρ mean: 0.59; range: 0.47–0.65, circles in **Fig 2D**) than those for different MEKi (ρ mean: 0.55; range: 0.46–0.64, triangles) (U test, $p = 0.10$). Overall, our results are consistent with those previously reported that response to anti-cancer drugs can be successfully predicted based on pan-cancer datasets [8,17,18,23,31]. Importantly, our framework provided strong *in silico* validation of the models' accurate cross-dataset predictions, as MEKi response and RNA-DNA data in the Klijn 2015 and CCLE datasets were collected independently.

## Between- and within-tissue performance of pan-cancer MEKi response predictions

To evaluate between-tissue performance of the pan-cancer predictions, we averaged the observed and predicted values for cell lines within each cancer type and asked if the model

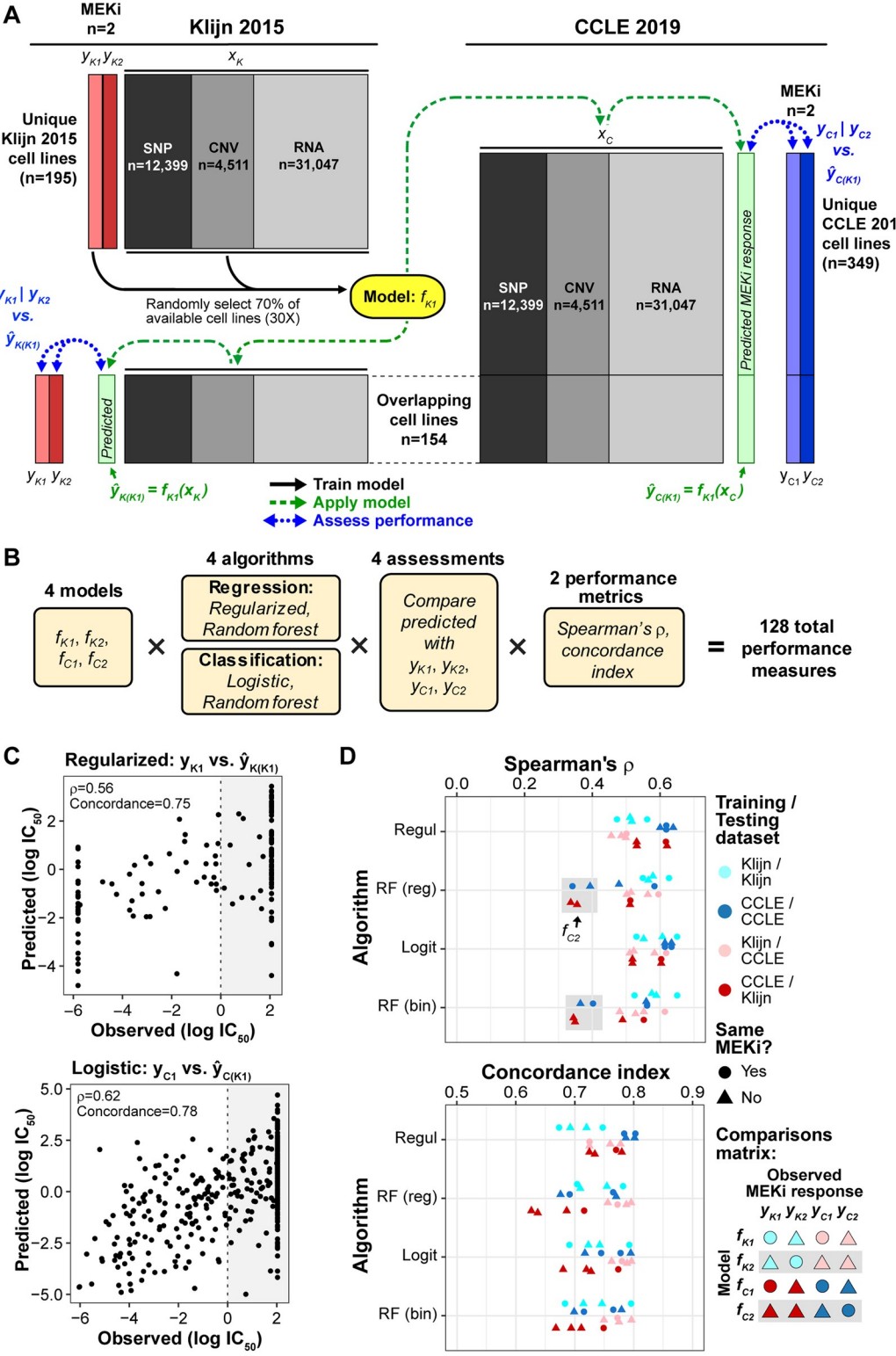

**Fig 2. Pan-cancer machine learning predictions of MEKi response. (A)** Schematic of an example of training-prediction-assessment workflow, depicting the generation of a prediction model (yellow, $f_{K1}$) that considers MEKi PD-901 response data from the Klijn 2015 dataset ($y_{K1}$, light red) and DNA and RNA features ($x_K$). The 154 cell lines in common between the two datasets were excluded from model building. Prediction models were built on 70% of training cell lines (selected randomly), repeated 30 times, and the final predicted drug response of a given cell line in the validation sets was calculated

as the average of the 30 repeats. The resulting prediction models are applied to within-dataset and cross-dataset RNA and DNA data ($x_K$ and $x_C$) to generate predicted drug response scores ($\hat{y}_{K(K1)}$ and $\hat{y}_{C(K1)}$). Predicted drug response values, shown in light green boxes, were then compared with observed drug response to evaluate model performance (within-dataset: $\hat{y}_{K(K1)}$ vs. $y_{K1} | y_{K2}$; cross-dataset: $\hat{y}_{C(K1)}$ vs. $y_{C1} | y_{C2}$). Model generation is depicted with black arrows, model application with green dashed arrows, and performance assessment with blue dotted arrows. **(B)** Outline of the full combinations of 4 models based on input data, 4 algorithms, assessments by comparing predicted MEKi response to the 4 series of observed response data, and 2 performance metrics. **(C)** Two examples showing observed and predicted log(IC$_{50}$) from the $f_{K1}$ model: regularized regression and within-dataset validation (top panel) or logistic regression and cross-dataset validation (bottom). Rank correlation (Spearman's ρ) and concordance index are shown in the top left corner. **(D)** Performance of all combinations of models, algorithms (y-axis), and assessments by rank correlation (Spearman's ρ, top panel) and concordance index (bottom). Within-dataset performances are indicated by shades of blue: cyan/dark blue, while between-dataset performances are indicated by shades of red: pink/dark red. Models trained from CCLE data are indicated by the darker shade. **Gray boxes**: random forest models trained on CCLE-Selumetinib data ($f_{C2}$). **Regul**: regularized regression; **RF (reg):** regression-based random forest; **Logit:** logistic regression; **RF (bin):** classification-based (binary) random forest.

recapitulated the group-level differences across the 10 cancer types. The mean observed and mean predicted log(IC$_{50}$) values were significantly correlated across cancer types (ρ range: 0.88–0.98; $p < 0.002$ using *cor.test* in R for all four comparisons shown in **Fig 3**), indicating that the pan-cancer prediction models accurately captured the average drug responses of tissues. Next, to evaluate within-tissue performance, we calculated the correlation between the observed and predicted log(IC$_{50}$) values from the pan-cancer predictions for cell lines *within* a single cancer type, and depicted the strength of correlation as the shape and tilt of the ellipsoid for individual cancer types (**Fig 3**). Based on hierarchical clustering of the resulting within-tissue prediction performances from different algorithms and training/testing sets (**Fig C** in S1 Text), we identify five tissues whose within-tissue variability was more accurately predicted by pan-cancer prediction models than other tissues (within-tissue ρ > 0.5): liver, ovary, lymphoid, stomach, and skin (mean within-tissue ρ = 0.59; ρ range: 0.51–0.64). The remaining five tissues (pancreas, brain, colorectal, breast, and lung) exhibited lower within-tissue prediction performance (mean ρ = 0.28; ρ range: 0.11–0.40). This result shows that whether a pan-cancer MEKi response prediction model is informative for an individual tumor (or patient) depends on which tissue is considered, with half of the tissues evaluated here achieving within-tissue predictions of ρ ≥ 0.51.

Accurate between-tissue predictions increase the overall prediction performance, which can be seen most clearly if we focus on two of the tissues, the brain and pancreas cell lines. Although within-tissue performance for either brain or pancreas cell lines was poor (both $r \leq 0.1$ for $y_{C1}$ vs. $\hat{y}_{C(K1)}$; **Fig 4A**; **Fig C** in S1 Text), prediction performance increased dramatically when the two tissues are considered together ($r = 0.58$; **Fig 4A**). This is because the two tissues are different in both drug response and molecular profiles (**Fig 1C and 1E**) and the performance across the two tissues in this example is driven by between-tissue differences.

To quantify the contribution of between-tissue signals on the performance of pan-cancer predictions, we standardized the observed and predicted MEKi response within tissues so that the per-tissue mean values are centered (an example in **Fig 4B**), and assessed the change in performance calculated from tissue-standardized values relative to the initial, non-tissue-standardized values (**Fig 4C**). Compared to the initial performance, the rank correlation coefficients using tissue-standardized observed and predicted log(IC$_{50}$) values were reduced from a range of 0.46–0.64 to 0.26–0.46 (paired U test, $p < 0.008$), depending on the algorithm and training/testing data, with all 8 pairs of training/testing data exhibiting reduced performance when observed and predicted MEKi response was standardized within tissue (**Fig 4C**). We contrasted the tissue-standardized predictions with those from a regularized regression model that considered only tissue-of-origin as features (n = 11 binary features; 10 tissues and an

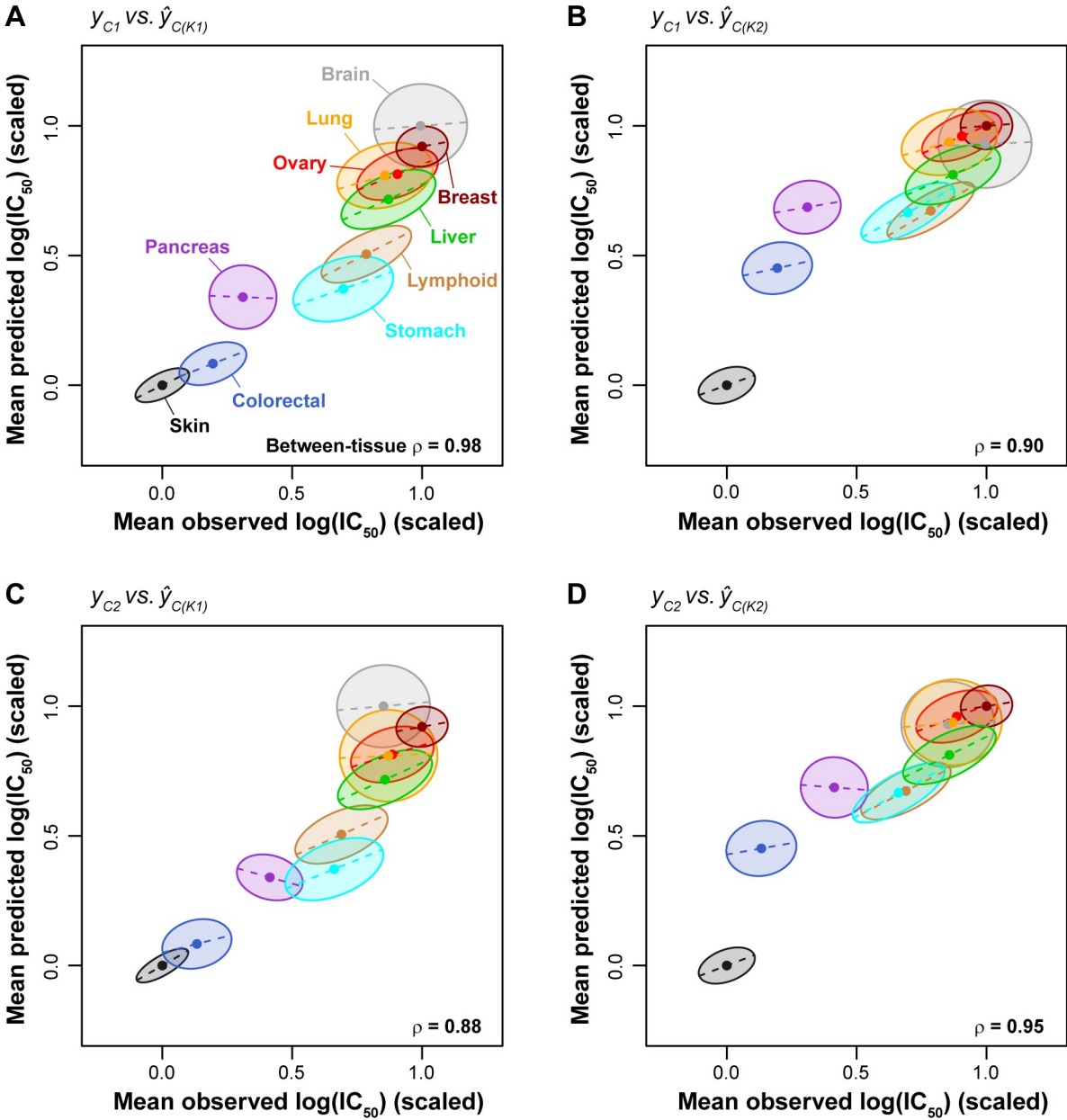

**Fig 3. Cigar plots of pan-cancer MEKi response predictions within and between cancer types.** Dots at the center of ellipses indicate the mean observed and predicted log(IC$_{50}$) values for a given tissue (mean values were scaled linearly between 0 and 1). ρ values indicate rank correlation among centers of ellipses. Maximum ellipsis length scales with the range of MEKi responses for a given tissue, where tissues with larger response ranges are associated with longer ellipses (e.g. stomach and lymphoid) while tissues with a smaller response range are associated with shorter ellipses (e.g. skin and breast). The slope of dashed lines (and tilt of ellipses) corresponds to the within-tissue regression coefficient of the predicted values against the observed values (Pearson's r). The width of ellipses also corresponds to the within-tissue regression coefficient, i.e., a high correlation value is shown as a slender ellipse while a low correlation value leads to a round ellipse. **(A-D)** Within- and between-tissue performances for the 4 combinations of drug/models trained on Klijn 2015 data and applied to CCLE data.

'other' designation; **Fig 1B**). Unsurprisingly, the predictions based on tissue alone performed worse than pan-cancer predictions that considered all molecular features (U test, $p < 4\text{x}10\text{-}4$; **Fig 4C**), but we also found that the performance of the tissue-only model was similar to the tissue-standardized pan-cancer predictions (U test, $p = 1$; **Fig 4C**).

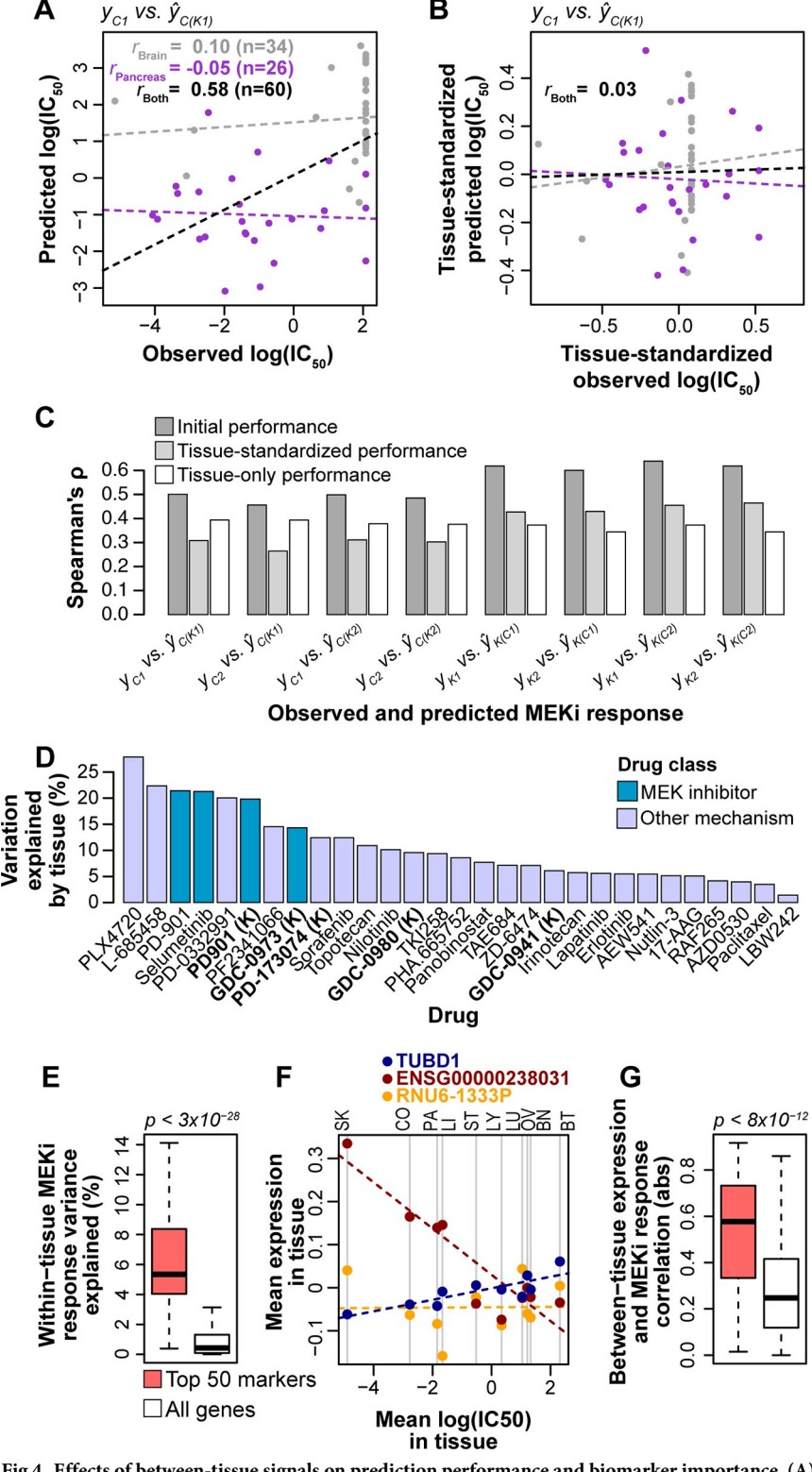

**Fig 4. Effects of between-tissue signals on prediction performance and biomarker importance.** (A) Performance of pan-cancer prediction models for a combination of brain and pancreas tissues. Correlation (Pearson's *r*) is shown for brain cell lines (gray), pancreas cell lines (purple), and both brain and pancreas combined (black). Dashed lines indicate lines of best fit. (B) Performance of pan-cancer prediction models for a combination of brain and pancreas tissues following standardization of observed and predicted MEKi response within tissues. Standardization was

performed by scaling linearly between 0 and 1 and subtracting the scaled mean. **(C)** Comparisons of initial performance (dark gray bars; **Fig 2D**), performances calculated following standardization of observed and predicted $log(IC_{50})$ values within tissues (light gray bars), and performances of tissue-only predictions (white bars) for regularized regression models. **(D)** Variation in drug response explained by tissue for the 29 drug screens in the Klijn 2015 (n = 5; bold and suffixed with "(K)" on x-axis) and CCLE 2019 (n = 24) datasets. Variation explained was calculated with analysis of variance (ANOVA) on a linear model of $log(IC_{50}) \sim tissue$. **(E)** Within-tissue variation in drug response explained by gene expression for the top 50 RNA biomarkers (pink) and all genes (white) for the $f_{K1}$ regularized regression model. *P*-value from a Mann-Whitney U test. **(F)** Example correlation between mean $log(IC_{50})$ and mean expression levels within tissues for three of the top 50 markers (red, blue, orange) for the $f_{K1}$ regularized regression model. Dashed lines indicate lines of best fit for the color-matched points. Vertical gray lines denote mean $log(IC_{50})$ values for the 10 tissues, which are abbreviated at the top of the plot: SK: skin, CO: colorectal, PA: pancreas, LI: liver, ST: stomach, LY: lymphoid, LU: lung, OV: ovary, BN: brain, BT: breast. **(G)** Absolute (abs) correlation between mean log(IC50) and mean gene expression across 10 tissues for the top 50 RNA biomarkers (pink) and all genes (white) for the $f_{K1}$ regularized regression model. *P*-value from a Mann Whitney U test.

We also developed regularized regression models that included both tissue-of-origin and RNA expression and DNA variant features. These models performed similarly to those that do not include tissue label (paired U test, *p* = 0.74), and they also exhibited a reduction in performance when observed and predicted drug responses were standardized within tissue (**Fig D** in S1 Text), with all 8 pairs of training/testing data exhibiting reduced performance when standardized by tissue (paired U test, *p* < 0.008). The MEK inhibitors in the Klijn 2015 and CCLE 2019 datasets are influenced to a greater degree by tissue-of-origin than most of the other 25 drugs (*p* < 0.02, U test; **Fig 4D**), although some drugs had even higher tissue effects than MEKi. This suggests that between-tissue effects for other drugs need to be examined individually.

At the feature level, we evaluated the within-tissue and between-tissue contributions to MEKi response predictions for all 31,047 RNA expression features, and then compared the top 50 biomarkers for each model—those having the highest regularized regression weights—to the remaining 30,097 features. To examine within-tissue feature contributions, we performed Type I Analysis of Variance (ANOVA) for each MEKi screen on each gene expression value based on a linear model: $y \sim tissue + gene expression$, where *y* represents the $log(IC_{50})$ for a given MEKi screen. We found that, as expected, the top 50 biomarkers explained a significantly higher amount of within-tissue MEKi response variation than other features (U tests, all $p < 4 \times 10^{-11}$; **Fig 4E**; **Fig E** in S1 Text). We next examined the between-tissue signals carried by the top 50 biomarkers by calculating, for each gene, the correlation between mean $log(IC_{50})$ and mean gene expression across tissues. These calculations are illustrated in **Fig 4F** for 3 examples from the top 50 biomarkers of $f_{K1}$ regularized models, including one example of positive correlation (TUBD1), one of negative correlation (ENSG00000238031), and one with no correlation (RNU6-1333P). The top 50 biomarkers for each model carry strong between-tissue signals, with significantly higher correlation between mean gene expression and mean MEKi response across 10 tissues (U tests, all $p < 9 \times 10^{-12}$; **Fig 4G**; **Fig E** in S1 Text). The feature weights and between- and within-tissue metrics for all 31,047 RNA expression features are provided for each model in **S1–S4 Datasets**. The overlap among top 50 biomarkers for different models are described in **Table B** in S1 Text with $f_{K1}$ and $f_{K2}$ sharing 15 biomarkers, $f_{C1}$ and $f_{C2}$ sharing 24 biomarkers, and 0–2 biomarkers shared for the four possible pairs of between-dataset models. Overall, these results indicate that between-tissue signals make a strong contribution to the performance of pan-cancer drug response predictions and markers most important to the models often contain both within-tissue and between-tissue signals.

## Sample size advantage of pan-cancer models over tissue-specific models

We next asked if pan-cancer prediction models can outperform those generated by considering a single cancer type (e.g. [13]). To address this question, we compared single-tissue

prediction models (i.e. trained and tested with cell lines from a single cancer type) and pan-cancer prediction models. As CCLE has a larger sample size, we considered only regularized regression models trained on CCLE and tested on Klijn 2015. We found that for the liver, lymphoid, and colorectal cancers, the performance of pan-cancer model is similar to that of single-tissue model; whereas for the other six cancer types: ovary, stomach, skin, lung, breast, and brain, the pan-cancer predictions are consistently more accurate (**Fig 5**). In all, the pan-cancer models performed better than or as well as tissue-specific models for 9 out of 10 cancers, whereas pancreas was the only cancer type for which tissue-specific models outperformed the full pan-cancer models.

A caveat to this result is that pan-cancer models were trained with 4- to 28-times more cell lines than tissue-specific models (e.g. 68 lung and 12 liver cell lines for tissue-specific models). To evaluate the effect of sample size we added an analysis where we down-sampled the pan-cancer data to the same sample size of the single-tissue models ("Pan-cancer (downsampled)" in **Fig 5**). These equal-sized pan-cancer models rarely outperformed tissue-specific models, except in cases where tissue-specific models perform particularly poorly (e.g. skin and lung cancers). Thus, the advantage of pan-cancer prediction models is only apparent when they are based on a larger sample size than tissue-specific models. At comparable sample size, tissue-specific models tend to be more accurate.

### Estimating sample sizes required for optimal prediction performance

The analysis above raises the practical question: how many cell lines are needed to produce optimal performance for pan-cancer models? To assess the relationship between prediction performance and sample size, we developed regularized regression models from a series of randomly downsampled sets of pan-cancer cell lines ($n_{cell\ lines}$ = 20–300; step size = 10; $n_{iterations/step}$ = 30). For this analysis, we developed only downsampled *pan-cancer* prediction models, and assessed performance in both pan-cancer and tissue-specific testing sets. For the tissue-specific testing subsets, we focused on cell lines from liver, ovary, lymphoid, stomach, and skin tissue, as these five tissues were well-predicted by pan-cancer prediction models (as shown in **Fig 3**; **Fig C** in S1 Text). For overall pan-cancer prediction, we observed an inflection point for training sample size between 70 and 100 cell lines (~70 cell lines for $f_{C1}$, **Fig 6A and 6B**; ~100 cell lines for $f_{C1}$, **Fig 6C and 6D**). Including additional cell lines further increased performance, but with diminished rate of improvement. Tissue-specific predictions by pan-cancer models with >100 cell lines resulted in diminishing returns for lymphoid and stomach cell lines, while the performance reached saturation even more quickly for liver and skin cell lines (**Fig 6**). However, performance for ovary cell lines showed no inflection point; it continued to increase as additional cell lines were included (**Fig 6**), suggesting ovarian cancers may be particularly well-positioned to benefit from even larger pan-cancer datasets. Overall, we find that generally 70–100 samples are needed to provide robust pan-cancer prediction performance in most cases.

### Discussion

The feasibility of pan-cancer drug response prediction is predicated on the existence of shared molecular vulnerabilities across cancer types. When successful, these predictions can contribute to the practice of precision oncology by informing individualized treatment decisions. In this study, by using pharmacogenomic datasets for MEK inhibition we confirmed that pan-cancer models did produce the level of performance (**Fig 2C**) reported in similar studies. However, unlike other studies, we underscore the observation that between-tissue differences make a substantial contribution to the apparent performance of pan-cancer models. This is because

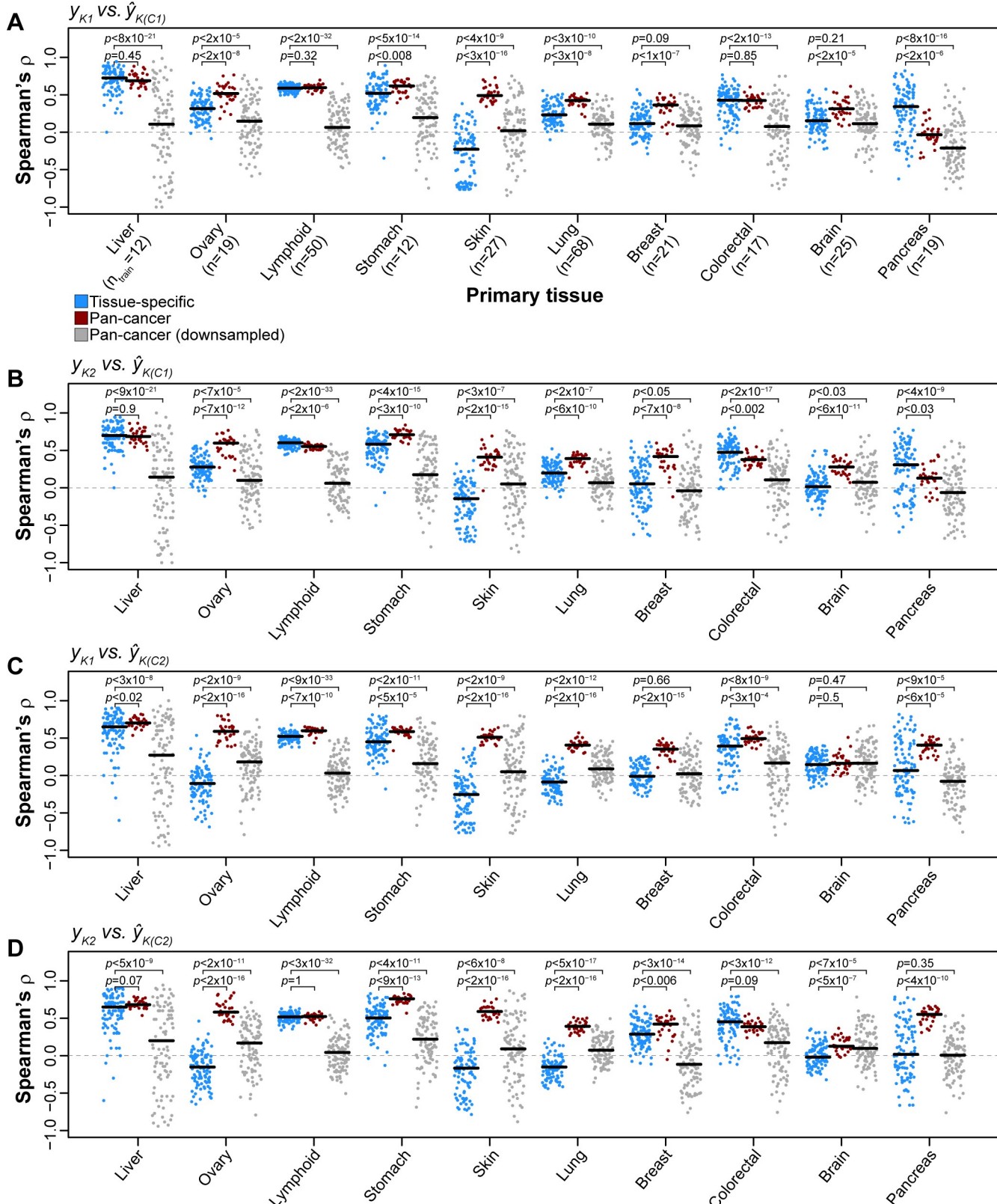

**Fig 5. Comparisons of pan-cancer and tissue-specific MEKi response prediction models.** For each tissue with ≥15 cell lines in both Klijn 2015 and CCLE datasets, a tissue-specific prediction model was trained and tested by considering only cell lines from a given tissue. Regularized regression prediction models

were trained on a random selection of 75% of cell lines in CCLE data of a given tissue type and applied to Klijn 2015 cell lines of the same tissue type (repeated 100 times). Performance was reported using rank correlation between observed and predicted MEKi drug response for each iteration (blue points). Rank correlation for the 30 iterations of MEKi response predictions based on pan-cancer prediction models (Fig 2) for a given tissue are indicated with red points. Pan-cancer prediction models were trained and tested using many more cell lines than tissue-specific prediction models. A new set of pan-cancer prediction models were generated by downsampling the available pan-cancer cell line sets to sample sizes equal to tissue-specific prediction models (gray points). (A-D) Results are shown for the 4 combinations of drug-models trained using CCLE data and applied to Klijn 2015 data. **Black horizontal lines**: median performance for a given distribution. Sample sizes on the x-axis in (A) indicate the number of cell lines used to train tissue-specific and downsampled pan-cancer models for the associated tissue. *P*-values from Mann-Whitney U tests.

many cancer types differ in both molecular features and drug response. The effects of between-tissue signals on the prediction models are reflected in the markedly decreased performance when only within-tissue effects are examined (Fig 4). Similarly, biomarkers most important to the prediction models tend to carry both within- and between-tissue signals (Fig 4E-4G; in S1 Text). As datasets in genome science typically contain multiple levels of heterogeneity, assessing the contributions of between- and within-group signals in prediction models represents an issue that extends beyond pan-cancer drug response predictions. We also find that pan-cancer models can outperform tissue-specific models, although this advantage usually disappears at comparable sample sizes (Fig 5). In practice, when the sample size for an individual cancer type is small, pan-cancer models can be developed from a larger, mixed-type training set, and applied as a powerful tool for informing anti-cancer therapies. The benefit is especially pronounced for the cancer types that contain a wide within-type variation of responses (e.g. MEK inhibition drugs in lung, liver, and stomach cancers).

While our analyses are limited to two publicly available datasets that contain gene expression, SNP, and CNV data, we can draw several lessons. First, the overall prediction performance across all cancer types include both between-tissue and within-tissue effects, which is consistent with previous results indicating characteristics of tissues are a major contributor to pan-cancer drug response predictions [14]. Additional assessment and reporting for individual cancer types is recommended. Pan-cancer drug response predictions can also incorporate additional data types, such as methylation and proteomics data. We expect that prediction performances based on data types that also vary with tissue-of-origin will be similarly affected by both between- and within-tissue signals. Second, prediction performance is variable between cancers of different tissue origin. As a result, a pan-cancer prediction score should be used with greater caution for an under-performing drug-tissue setting (e.g., MEK inhibitor response in pancreas, brain, or colorectal cancers). In practice, the prediction score for a given tumor may come with a claim of high accuracy if such scores are evaluated in a pan-cancer setting, but for the individual variability within the tumor type it may offer little predictive power. In this situation, using the prediction score would be akin to treating the tumor as an average sample of that tumor type, thus not much different from making a decision based on the tumor type. Third, biomarkers identified using pan-cancer models can emerge for different reasons: some driving a strong response in one of a few tissues, others involved in drug-response programs shared in common by many more tissues, still others could simply be tissue-specific genes not involved in drug-response. Marker selection in the future will need to consider the diversity of tissue origin in the training panel as well as the tissue type of the application.

We also highlight three observations that are relevant to designing future experiments or investigating additional cancer models and drug families. First, when assessing sample size of pan-cancer models for MEKi, we show that inclusion of more than 100 cell lines frequently led to diminishing performance returns. However, more or fewer cell lines will likely be required to saturate prediction performances based on the number of cell lines present for each tissue, as well as the similarity between the tissue groups. Second, pan-cancer models for one MEK

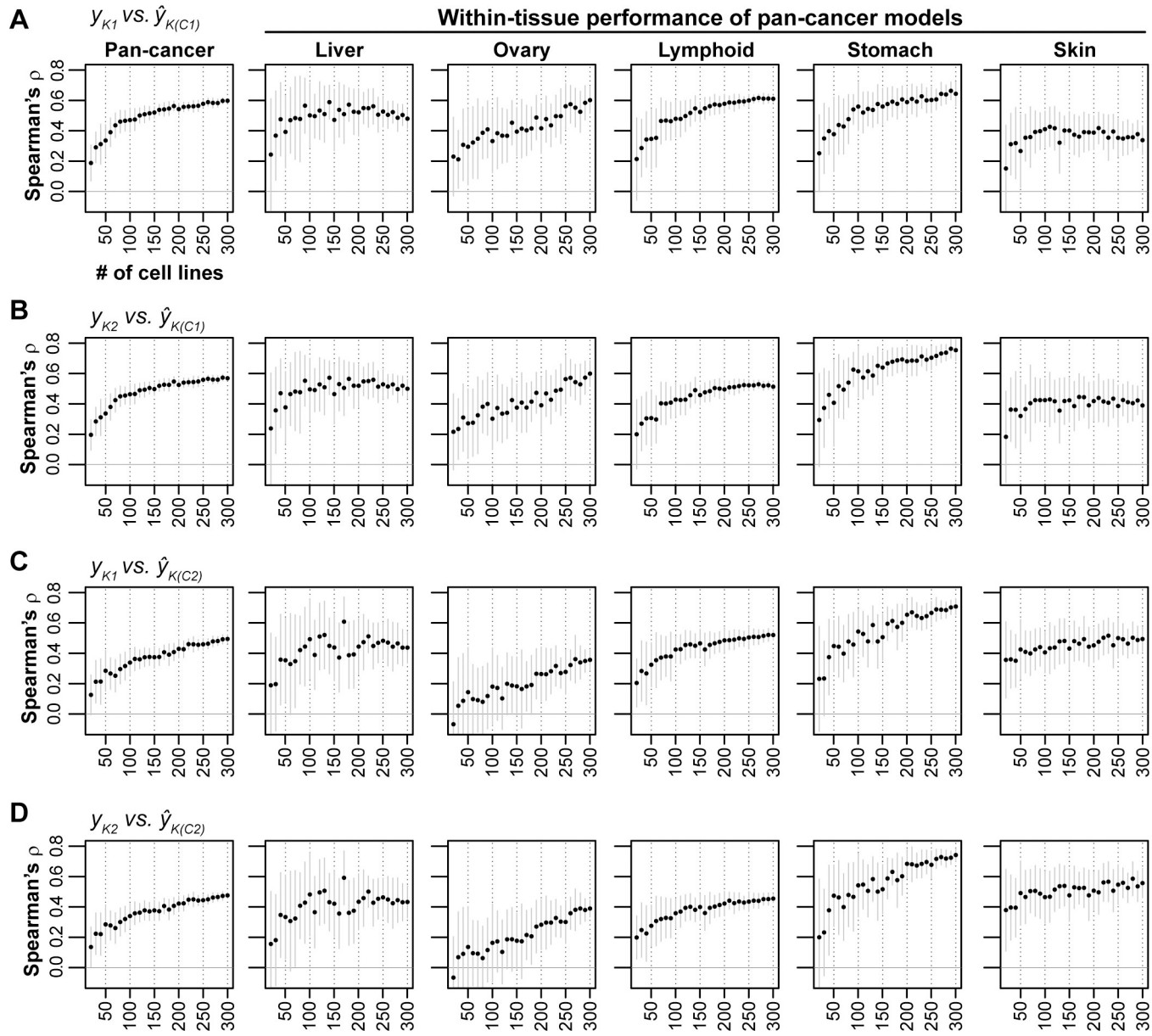

**Fig 6. Performance of pan-cancer MEKi response predictions based on downsampled sets of cell lines.** Downsampled pan-cancer models were applied, from left to right, on pan-cancer testing sets and five single-tissue testing sets. Cell lines in the CCLE dataset were randomly downsampled to sample sizes ranging from 20 to 300 in 10 cell line increments (x-axis; $n_{iterations}$ = 30). Regularized regression prediction models were trained on downsampled cell lines sets and applied to cell lines in the Klijn 2015 dataset. Mean ranked correlation (Spearman's ρ) between observed and predicted logged $IC_{50}$ values across the 30 iterations for each sample size are plotted. **(A-D)** Results are shown for the 4 combinations of drug-models trained using CCLE data and applied to Klijn 2015 data. **Solid gray vertical lines**: ±standard deviation.

inhibitor (e.g. PD-901) led to successful predictions for other MEK inhibitors (e.g. Selumetinib; **Fig 2**). The MEK inhibitors considered in this study share a specific mechanism of action: potent inhibition of the MEK1 protein (although PD-901 also inhibits MEK2) [32]. Third, across the ten cancer types, although lymphoid cell lines are biologically and clinically distinct from the other, epithelial-origin cell lines (see **Fig 1E**), the pan-cancer models can accurately predict this cancer type (**Fig 3**; **Fig C** in S1 Text). This suggests lymphoid cell lines may respond to MEK inhibition via similar biological pathways.

With the ever-increasing availability of large pharmacogenomic datasets and the introduction of basket clinical trials, which select patients based on tumor features rather than cancer type, pan-cancer prediction models and biomarker sets will be increasingly adopted to match patients with effective treatments. Dissecting between-tissue and within-tissue statistical signals in drug response prediction models is essential for contextualizing "precision" in precision oncology, allowing the balanced consideration of group-based (i.e. tissue class) treatment decisions on one hand, and individual-based decisions on the other. In addition, deep learning approaches to drug response predictions [23,24] will become increasingly promising as pharmacogenomic datasets continue to grow, as these approaches frequently lead to increased predictive performance, but require large datasets to unlock their full potential.

Our examination of between- and within-tissue signals in pan-cancer drug response prediction models echoes similar concerns in other fields that involve joint contributions of individual and group factors. For example, shared ancestry is a known confounder in genomewide association studies, where correlations between traits and ancestry can lead to false positives in association tests and mis-estimates of marker effects, as has been demonstrated for variation in anthropomorphic traits (e.g. height) or disease risks (e.g. Type 2 diabetes) [33–37]. Similarly, forensic markers chosen based on their ability to identify individuals also carry population-level information [38]. Plant and animal breeding approaches that utilize genomewide SNP patterns to predict economically-important traits, known as genomic selection, must be adjusted to ensure informative markers explain variation in the trait of interest, rather than population structure [39,40]. Ultimately, developing an understanding of both the group-level and individual-level factors affecting individual trait values is needed in interpreting prediction models and marker panels, especially those generated from highly heterogeneous populations.

## Methods

### Cell line annotations and drug response data

Drug response and molecular characterization data were retrieved from two sources: 1) the supplemental material of a prior publication (Klijn 2015) [11] and 2) a tumor cell line database (CCLE) [12]. For Klijn 2015, cell line annotations, including identifiers and primary tumor types, were retrieved from "Supplementary Table 1" in [11]. Annotations for CCLE were retrieved from "Supplementary Table 1" in [12]. Cell line identifiers were cross-mapped across datasets and data modalities (i.e. drug response, RNA expression, and DNA variants) by accounting for differential use of upper/lower case characters and/or non-alphanumeric characters; alphabet characters in cell line identifiers were converted to upper case and identifiers were stripped of non-alphanumeric characters. Klijn et al [11] excluded 65 cell lines from analysis due to SNP similarity to other cell lines or uncertainty in cell line origin; we also excluded these cell lines from our analyses ("In Final Analysis Set" column = FALSE; "Supplementary Table 1" in [11]).

Klijn 2015 tumor cell lines were screened with 3–4 replicates across 9 drug doses (range: 0.15 nM to 20 μM); while the CCLE cell lines were screened with at least 2 replicates across 8 drug doses (range: 2.5 nM to 8 μM). The resulting dose response curves were summarized with the $IC_{50}$ value: the dose at which 50% of cells are non-viable; lower $IC_{50}$ values indicate greater sensitivity. $IC_{50}$ can be difficult to interpret, as it does not account for different shapes in dose response curves or for differences in minimum viable cells observed. However, $IC_{50}$ was used in this study as it was the sole drug response measure in common between the Klijn 2015 and CCLE 2019 datasets. For Klijn 2015, we retrieved drug response data from "Supplementary Table 13" in [11]. Among the 5 drugs available in Klijn 2015, we focused on two MEK

inhibitors: PD-0325901 (referred to as PD-901) and GDC-0973. For CCLE, drug response data were downloaded in the file "CCLE_NP24.2009_Drug_data_2015.02.24.csv" (database file date: 24 February 2015), which included $IC_{50}$ data for 24 anti-cancer drugs. We used drug response data for two MEK inhibitors in CCLE: PD-901 and AZD-6244 (trade name: Selumetinib). The natural log of $IC_{50}$ values were used for feature selection, parameter sweeps, and model training and evaluation.

## RNA expression data

For Klijn 2015, we analyzed expression data from RNA-seq available from two supplemental datasets, designated as protein-coding ("Supplementary Data 1" in [11]) and non-coding ("Supplementary Data 2" in [11]). In all, there were 25,984 coding genes and 21,398 non-coding genes after excluding genes that were invariable in expression across all 675 cell lines with RNA data. Expression levels were provided as variance stabilization-transformed read counts from the DESeq package [41]. We further standardized RNA expression data for each gene by linearly scaling values across cell lines to a range between 0 and 1 and shifting the scaled values by subtracting the scaled mean. Standardization was performed so that genes are present in similar scales and thereby contribute comparably to prediction models. In addition, standardization brings the RNA data from Klijn 2015 and CCLE 2019 datasets into similar ranges, which allowed a model from one dataset to be effectively applied to the second dataset. Gene identifiers in the protein-coding table were converted from Entrez Gene ID to ENSEMBL format using the org.Hs.eg.db package [42] in R, while gene identifiers in the non-coding table were already provided in ENSEMBL format. Once gene identifiers for the Klijn 2015 protein-coding and non-coding datasets were both in ENSEMBL format, we removed genes from the non-coding dataset that were found in the coding dataset. Following processing described above, the coding and non-coding expression data were merged and treated as a single dataset for downstream analysis.

For CCLE, we retrieved expression data from RNA-seq analysis as a single table from the CCLE database (file name: "CCLE_RNAseq_genes_rpkm_20180929.txt"; database file date: 29 September 2018). Expression levels for 55,414 genes (after excluding genes that were invariable in expression across all 1,019 CCLE cell lines) were provided as reads per kilobase of transcript per million mapped reads (RPKM). The natural log of CCLE RNA values was taken by log (RPKM+0.001), as the data exhibited a long right tail with a small number of extremely high expression values. CCLE gene identifiers for RNA expression data were provided in ENSEMBL format. CCLE RNA expression data was standardized as described above for the Klijn 2015 dataset (i.e. scaling and shifting). Standardization was performed independently for both Klijn 2015 and CCLE datasets. For downstream analysis (e.g. feature selection, model building), we included only the 31,047 genes with RNA expression data present in both Klijn 2015 and CCLE RNA-seq datasets, while genes present in only one dataset or the other were excluded.

## DNA variant data

For Klijn 2015, we retrieved single nucleotide polymorphism (SNP) and copy number variation (CNV) mutation data from the associated supplementary material (SNPs: "Supplementary Data 3" in [11]; CNVs: "Supplementary Data 4" in [11]). Klijn 2015 SNP and CNV data were identified from SNP microarray analysis (Illumina HumanOmni2.5_4v1). The provided SNPs had been previously filtered to include only missense and nonsense mutations. In all, there were 127,579 unique SNPs across 14,375 genes for all 675 cell lines with SNP calls. Because any given SNP was rare in the Klijn 2015 dataset, SNPs were mapped up to the gene level; genes then served as binary features (i.e. gene mutated / gene not mutated) without regard to which

SNP(s) a gene contained. For Klijn 2015, CNV data were provided on a gene-by-gene basis as ploidy-adjusted copy number calls from the SNP array data using the PICNIC [43] and GIS-TIC [44] algorithms. We then classified the provided ploidy-adjusted copy number values as amplifications (coded as 1) or deletions (coded as -1) based on thresholds described by Klijn et al [11] (amplifications: ≥1; deletions: ≤-0.75). In total, 18,396 gene amplifications and 42,371 gene deletions for 11,864 unique genes were present across 668 cell lines with CNV calls. Gene identifiers for both the SNP and CNV data in Klijn 2015 were converted from Entrez Gene ID to ENSEMBL format using the org.Hs.eg.db package [42].

For CCLE, we retrieved whole exome sequencing variant calling data from the CCLE database (file name: "CCLE_mutation_DepMap_18q3_maf_20180718.txt"; database file date: 18 July 2018). Of the 1,203,976 total variant calls present, only the 705,056 single-nucleotide variants annotated as either missense or nonsense mutations were included in further analysis–consistent with the SNPs available in the Klijn 2015 dataset. As described above for the Klijn 2015 dataset, SNPs were mapped up to the gene level where the mutation status of a gene served as a binary feature. The CCLE dataset contains more than 5 times as many missense and nonsense SNP calls than the Klijn 2015 dataset, which is largely due to the different platforms employed for genotyping (i.e. SNP array vs. exome-seq). The effects on downstream analysis brought by differences in the total number of SNP calls between the two datasets is tempered by mapping of SNPs to the gene level. Gene identifiers were converted from Entrez Gene ID to ENSEMBL format using the org.Hs.eg.db package [42]. CCLE CNV data were downloaded from the CCLE database (file name: "CCLE_MUT_CNA_AMP_DEL_binary_Revealer.gct"; database file date: 29 February 2016). CNVs were provided based on calls from SNP microarray data (Affymetrix Genome-Wide Human SNP Array 6.0) and the GISTIC algorithm [44]. The downloaded CNV file provided gene-based amplification and deletion calls (classification thresholds: amplifications: ≥0.7 ploidy-adjusted copy number; deletions: ≤-0.7), which were assembled into a gene-by-cell line matrix with amplifications coded as 1, deletions coded as -1, and no copy number variation coded as 0. In total, 691,866 gene amplifications and 948,159 gene deletions were called across 1,030 cell lines. Gene symbols in the CCLE CNV data were converted to ENSEMBL IDs using the org.Hs.eg.db package [42]. In total, 12,399 genes mutated by SNP and 4,511 genes with CNVs were in common between the two datasets and included for downstream analysis. This analysis did not attempt to distinguish germline variants from somatic variants, mainly because there were no genotype data available to serve as the matched "normal" for the cell lines. The processed drug response, RNA expression, point mutation, and copy number variation data as described above are available as R objects (RDS files) through CyVerse Data Store: https://de.cyverse.org/dl/d/43AB0125-4826-4599-9337-E8B61F41DBA4/lloyd_etal_pancancer_MEKi_processed_data.zip.

## Dimensionality reduction analysis

We performed dimensionality reduction by $t$-distributed stochastic neighbor embedding (t-SNE) and principal component analysis (PCA) on a subset of genes with highly variable expression patterns. For each gene, the variance and mean were calculated for distributions of the pre-logged and pre-standardized expression values. A measure of expression variability was then calculated as the log of the ratio of variance over mean (values calculated in the previous step). We ranked expression variability of each gene based on log variance-to-mean ratio and included the union of the 3,000 most variable genes from the Klijn 2015 and CCLE datasets in dimensionality reduction analysis shown in **Fig 1E** (t-SNE) and **Fig A** in S1 Text (PCA). The Rtsne package [45] was used to run t-SNE analysis (initial_dims = 10, theta = 0.25, all other parameters default).

## Cigar plots

We developed a visualization strategy to simultaneously display within- and between-group prediction performances, called the "cigar plot" (**Fig 3**). The cigar plot displays the predicted and observed drug response in an x-y plot, and highlights their concordance (i.e., the prediction performance) at two levels. First, the variation within each group is shown as an ellipse, with the within-group concordance (e.g., measured by rank correlation coefficients) shown as the tilt and width of the ellipses. Second, the between-group variation (differences of group mean values) is shown as the location of ellipsis centers. Ellipsis tilt was calculated by 45˚ × correlation coefficient: coefficient of 1 is associated with a 45˚ tilt, coefficient of 0 a 0˚ tilt, and coefficient of -1 a -45˚ tilt. The slope of the line across an ellipse is the same as ellipsis tilt. In **Fig 3**, correlation coefficients (Pearson's $r$) were used. Ellipsis length is scaled to the observed within-group range, assessed on a per-tissue basis by concatenating the 4 sets of MEKi responses in both datasets and calculating the interquartile range across all MEKi response values. Ellipsis width scales with within-group performance and is calculated as: ellipsis length × (1—absolute correlation coefficient). Larger absolute coefficients result in narrower ellipses (coefficient of 1 results in a line) while smaller coefficients result in rounder ellipses (coefficient of 0 results in a circle). R code to generate cigar plots is available at: https://github.com/johnplloyd/cigar_plot.

## Model building approach

We developed prediction models for four sets of input data ("4 models" in **Fig 2A and 2B**), trained by considering drug response data for the two MEK inhibitors in the two datasets as the outcome, with the associated molecular features as the independent variables–e.g. the response to PD-901 in Klijn 2015, to be predicted with the Klijn 2015 molecular features (**Fig 2A**). The two models from Klijn 2015 were denoted $f_{K1}, f_{K2}$, likewise the two from CCLE were denoted $f_{C1}, f_{C2}$. Within each model, we applied four algorithms: regularized regression, random forest (regression), logistic regression, and random forest (classification) (additional details below). In all, we trained 4 MEKi × 4 algorithms = 16 prediction models. To assess performance, we performed two types of validation: 1) within-dataset validation: training a model in a dataset using the non-shared cell lines (i.e. unique to a given dataset), predicting the drug response for the other, 154 shared cell lines using their molecular features in this same dataset, and compared with their actual drug response; 2) between-dataset validation: training a model in the same way as above (i.e., using the non-shared cell lines), predicting the drug response for all cell lines (both shared and non-shared) in the other dataset, using the molecular features in the other dataset, and compare with the actual drug response in the other dataset. An example schematic of model training while considering PD-901 in the Klijn 2015 dataset and model application both within and across datasets is shown in **Fig 2A**.

During the training phase, for regression-based algorithms, 30 prediction models were built after randomly-sampling 70% of available training cell lines, and the predicted validation set response was calculated as the average of the 30 randomly-sampled runs. This procedure is conceptually similar to a bootstrap aggregating (bagging) approach, but differs in that each of the 30 random training sets was sampled *without* replacement (whereas bagging methods are defined by sampling *with* replacement). Note that the remaining 30% of training cell lines that were not selected for each of the 30 iterations of model training were not used to assess model performance. For classification-based algorithms, 30 prediction models were trained using an equal number of sensitive and resistant cell lines: 70% of cell lines of the least populated class (typically sensitive) in the training set were randomly selected and matched with an equal number of randomly-selected cell lines of the more populated class (typically resistant). As

with regression-based algorithms, classification-based prediction models were applied to within-dataset and cross-dataset validation sets and the final prediction score was calculated as the average of 30 prediction models.

As mentioned above, we established sets of regularized regression, random forest (regression), logistic regression, and random forest (classification) prediction models. Regularized regression and random forest (for both regression and classification) models were trained using the *glmnet* [46] and *randomForest* [47] packages in R, respectively. Logistic regression models were trained using the lm() function (family = binomial) in the base installation of R. Regression prediction models (regularized and random forest) were trained on log($IC_{50}$) of MEKi response. Distributions of drug response by $IC_{50}$ were bimodal for all four MEKi screens (**Fig B** in S1 Text). As a result, we tested classification-based algorithms trained with cell lines binarized as sensitive or resistant based on $IC_{50}$ thresholds alongside regression-based algorithms that considered log $IC_{50}$ values directly. The threshold to binarize cell lines as sensitive or resistant to a drug was set at 1 μM, a value that readily split the bimodal $IC_{50}$ distributions for 3 of 4 MEKi screens and has been used previously to define cell lines as sensitive to MEK inhibition [48]. For the 4th MEKi screen, GDC-0973 in Klijn 2015, an alternative threshold, 6 μM, better separated the $IC_{50}$ distribution. For the GDC-0973 screen in Klijn 2015, we generated classification-based prediction models using both 1 μM and 6 μM $IC_{50}$ thresholds. We found that models trained using the 6 μM threshold performed similarly or worse than those using a threshold of 1 μM (**Fig B** in S1 Text), depending on the MEKi screen predicted. We therefore opted to use the results from the 1 μM threshold classification prediction models for GDC-0973 throughout the study.

For regularized regression predictions, the α parameter controls whether a ridge regression (α = 0), elastic net (0 < α < 1), or least absolute shrinkage and selection operator (LASSO; α = 1) model is generated and the λ parameter controls the strength of the penalty on model β values. We tuned the α and λ parameters through a parameter sweep. We tested α values from 0 to 1 in 0.1 step increments, while the λ values tested were 0.001, 0.01, 0.1, 1, and 10. For logistic regression, we performed feature selection by LASSO prior to model training. For the LASSO feature selection, we performed a parameter sweep to test multiple λ values: $1 \times 10^{-5}$, $5 \times 10^{-5}$, $1 \times 10^{-4}$, $5 \times 10^{-4}$, 0.001, 0.005, 0.01, 0.05, and 0.1. For random forest models (both regression and binary), we tuned three parameters: number of trees (ntrees parameter in randomForest package; values tested = 100, 250, 500, 750, and 1,000), number of features randomly-selected at each tree split (mtry; values tested = 218, 479, 4795, 11989, and 23978), and minimum node size in tree leaves (nodesize; values tested = 1, 5, 10, 25). Number of features tested were selected based on square root, 1%, 10%, 25%, and 50% of total features. Minimum node size affects tree size and depth, with larger values producing smaller trees. Parameter tuning was performed by randomly selecting 70% of training cell lines and applying models trained with different parameter sets to the remaining 30% of training cell lines, repeated for 30 iterations. Note that the validation cell lines were not used for parameter tuning. Optimal parameter sets were selected based on maximum mean performance in the testing set across the 30 iterations. Performance was measured by Spearman's ρ for regression algorithms and AUC-ROC for classification algorithms. **Table C** in S1 Text shows the optimal parameters selected for regularized and logistic regression, and **Table D** in S1 Text shows the optimal parameters for random forest. The R scripts used to run parameter sweeps are available on GitHub: https://github.com/johnplloyd/R_prediction_model_building.

We ran an additional parameter sweep for the $f_{C1}$ and $f_{C2}$ regularized regression models because the λ value selected for both models was the maximum value available. Using the same framework for parameter sweeps described in the previous paragraph, we tested λ values of 1, 10 (initial optimal parameter), 100, 1,000, and 10,000 and α values of 0 (initial optimal

parameter), 0.1, and 0.2. Based on this extended parameter sweep, we found that $\lambda = 10$ and $\alpha = 0$ remained as the optimal parameter set for the $f_{C2}$ regularized regression model. However, we found that the optimal parameter set selected for $f_{C1}$ from the second parameter sweep round was $\lambda = 1$ and $\alpha = 0$, which was a parameter set available in the initial sweep. We expect that there are likely multiple sets of parameters that provide similar, near-optimal performance, and that the final optimal set is influenced by the random selection of training instances during the parameter sweep model building. As a result, we have opted to maintain the parameters $\lambda = 10$ and $\alpha = 0$ for the $f_{C1}$ model that were selected in the first parameter sweep round.

For tissue-specific models, we trained the prediction models with CCLE data, randomly selecting 75% of cell lines of a given cancer type and applied the models to Klijn 2015 data for testing. If a cell line overlapping between Klijn 2015 and CCLE was selected to build the tissue-specific model, it was excluded from model testing. Tissue-specific prediction models were the only instance in which the 154 cell lines in common between Klijn 2015 and CCLE were used for training, as there were fewer cell lines available for individual cancer types. Parameters selected from pan-cancer prediction models (**Table C** in S1 Text) were also used for generating tissue-specific models.

## Supporting information

**S1 Dataset. Feature weights for regularized regression $f_{K1}$ model.**
(XLSX)

**S2 Dataset. Feature weights for regularized regression $f_{K2}$ model.**
(XLSX)

**S3 Dataset. Feature weights for regularized regression $f_{C1}$ model.**
(XLSX)

**S4 Dataset. Feature weights for regularized regression $f_{C2}$ model.**
(XLSX)

**S1 Text. Fig A.** Within- and between-tissue RNA expression similarity. **(A, B)** Scatterplots of the first two dimensions from principal component (PC) analysis on transcriptome data. Each point represents a cell line and is colored by dataset **(A)** and tissue-of-origin **(B)**. **(C,D,E)** Heatmaps of within- and between-tissue RNA expression similarity. Heatmap colors indicate the mean ranked RNA expression correlation (Spearman's $\rho$) across genes with highly-variable expression (see Methods) for 100 randomly-selected pairs of cell lines of the indicated tissues. Black outlines indicate comparisons within the same tissue type. Expression correlation is shown for cell lines **(C)** within Klijn 2015, **(D)** within CCLE, and **(E)** between the two datasets. **Fig B.** Evaluating thresholds to classify cell lines as sensitive or resistant to four MEK inhibitor screens. **(A-D)** $IC_{50}$ distributions for the four MEKi screens in Klijn 2015 (A, B) and CCLE 2019 (C, D). Vertical dotted lines: 1 μM threshold used to classify cell lines as sensitive or resistant to MEK inhibition. Vertical dashed line in **(B)**: 6 μM threshold tested for GDC-0973 in Klijn 2015. **(E,F)** Area under the receiver operating characteristic (auROC) performances calculated by comparing observed MEKi response (x-axis) with predicted MEKi response from $f_{K2}$ models using 1 μM (dark gray bars) or 6 μM (light gray) thresholds trained with logistic regression **(E)** or random forest **(F)** algorithms. **Fig C.** Prediction performances as assessed in the full pan-cancer cell line set (left-most column) and within 10 cancer types (the next 10 columns). Heatmaps indicate rank correlation (Spearman's $\rho$) between observed and predicted MEKi responses (**Fig 2**) based on regularized (top) and logistic (bottom) regression prediction

models. Each row is for a specific combination of training data and test data, over two MEK inhibitors and two datasets. Also shown are the mean performance for each column (bottom row). Dendrogram at the top depicts hierarchical clustering of the tissues by their performance patterns. **Fig D.** Comparisons of performance of regularized regression models that included tissue-of-origin as features alongside RNA expression and DNA variant features (dark gray bars) with performances calculated following standardization of observed and predicted log ($IC_{50}$) values within all tissues (light gray bars). Paired Mann-Whitney U test: $p < 0.008$. **Fig E.** Within- and between-tissue signals for the top 50 marker genes based on their maximum absolute regularized regression coefficient. (Left) Probability density distributions for a measure of within-tissue signal: the % of within-tissue variance in MEKi response explained for the top 50 markers (red) and all genes (gray). (Right) Probability density distributions for a measure of between-tissue signal: the absolute correlation ($r$) between the mean per-tissue gene expression and the mean per-tissue MEKi response. Vertical dotted lines indicate median values for color-matched distributions. *P*-values are from Mann-Whitney U tests comparing the two distributions in each panel. **(A-D)** Results are shown for the four MEKi screens. **Table A.** Alternative performance metrics for regularized and logistic regression algorithms. **Table B.** Overlap among the top 50 biomarkers from regularized regression models. **Table C.** Optimal model parameters for regularized and logistic regression algorithms. **Table D.** Optimal model parameters for random forest algorithms.
(DOCX)

## Author Contributions

**Conceptualization:** John P. Lloyd, Matthew B. Soellner, Sofia D. Merajver, Jun Z. Li.

**Data curation:** John P. Lloyd.

**Formal analysis:** John P. Lloyd.

**Funding acquisition:** John P. Lloyd, Sofia D. Merajver, Jun Z. Li.

**Investigation:** John P. Lloyd, Matthew B. Soellner.

**Methodology:** John P. Lloyd, Jun Z. Li.

**Project administration:** Sofia D. Merajver, Jun Z. Li.

**Resources:** Sofia D. Merajver, Jun Z. Li.

**Software:** John P. Lloyd, Jun Z. Li.

**Supervision:** Sofia D. Merajver, Jun Z. Li.

**Validation:** John P. Lloyd, Matthew B. Soellner, Sofia D. Merajver, Jun Z. Li.

**Visualization:** John P. Lloyd, Sofia D. Merajver, Jun Z. Li.

**Writing – original draft:** John P. Lloyd.

**Writing – review & editing:** John P. Lloyd, Matthew B. Soellner, Sofia D. Merajver, Jun Z. Li.

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
