## [Decision Letter · Decision Letter 0]

27 Jul 2020

Dear Dr. Li,

Thank you very much for submitting your manuscript "Impact of between-tissue differences on pan-cancer predictions of drug sensitivity" for consideration at PLOS Computational Biology.

As with all papers reviewed by the journal, your manuscript was reviewed by members of the editorial board and by several independent reviewers. In light of the reviews (below this email), we would like to invite the resubmission of a significantly-revised version that takes into account the reviewers' comments.

The paper has been commented by three Reviewers. The Reviewers found the paper interesting, but requiring improvements. Before submitting the revised version of the manuscript, please address their comments carefully.

We cannot make any decision about publication until we have seen the revised manuscript and your response to the reviewers' comments. Your revised manuscript is also likely to be sent to reviewers for further evaluation.

Sincerely,

Ewa Szczurek

Guest Editor

PLOS Computational Biology

Florian Markowetz

Deputy Editor

PLOS Computational Biology

The paper has been commented by three Reviewers. The Reviewers found the paper interesting, but requiring improvements. Before submitting the revised version of the manuscript, please address their comments carefully.

Reviewer's Responses to Questions

**Comments to the Authors:**

Reviewer #1: 1. There is a good focus on the literature review within the introduction, however, I recommend the authors to rewrite the introduction section, as some sections of it, is difficult to follow, specifically connections between biological concepts.

2. “The observation that cell lines respond similarly to different MEK inhibitors indicates cross-MEKi predictions are feasible, provided the two compounds are chemically similar.” That means those who have similar drug response have similar compound structure. Did you study if other way around is also valid? Meaning that if two compounds are similar then their drug responses are similar? Or with further analysis (i.e. stringing, REFINED, OmicsMapNet, and etc) that includes fingerprints for the compounds.

3. Authors have claimed that “Cell lines from the same primary tissue tend to be present in similar regions of the PC1-PC2 space (Figure 1E)”. It is nice to visualize in such a approach, however the conclusion is not necessarily correct, as only for one tissue we can confidently make such a conclusion not for the rest. That may be because of information loss that embedded in any dimensionality reduction technique such as PCA. To ensure that the conclusion is correct I urge the authors to consider applying other dimensionality reduction techniques such as MDS, Isomap, t-SNE, or so many others to evaluate if they can reach such a conclusion.

4. It was mentioned in the paper that “Cell lines from the same primary tissue tend to be present in similar regions of the PC1-PC2 space (Figure 1E) and have correlated RNA expression levels (Figure S1), highlighting that cell lines derived from the same primary tissue have similar transcriptomic features.” Seems like authors are considering correlation between 0.3-0.4 as high correlation. I don’t think that is a fair call, and the threshold must be larger value. On the other hand, correlation has its own problems that are well known in stat community. For instance, correlation between to signals (sin(x) and cos(x)) is zero, however we know that they are the exact same signal with some delay, which indicates correlation cannot capture some important information. For more accurate analysis, I recommend authors to consider other metrics such as Cosine similarity, R2 or mutual information, to measure similarity between RNA expressions.

5. Authors must explain how they pick the threshold value for dichotomizing the cell lines responses as sensitive and resistive.

6. Authors must report NRMSE, NMAE, and R2 for their predictive models, as one model can have high correlation value and high error value at the same time. On the other hand, the scatter plot of the models in figure 2C doesn’t show a good performance by the models. For instance, there are so many observed values of log(IC50) = 2 that are predicted ranging from -5 to 5.

7. How do you perform auROC for a regression task? Could you please show a reference for that? As far as I know auROC is only possible when you have probability value for a classification task.

8. Authors should mention which statistical test did they use for the “Between- and within-tissue performance of pan-cancer MEKi response predictions” section.

9. In the section where authors investigated “if pan-cancer prediction models can outperform those generated by considering a single cancer type.”, there are so many avenues for improvement as it is very interesting to perform such an investigation, however authors couldn’t achieve superior performance. To this end, I recommend authors to perform some other non-linear regression model such as Random forest, XGBoost and so on. Also perform robustness analysis similar to the community effort paper: https://www.nature.com/articles/nbt.2877

10. In the “Estimating sample sizes required for optimal prediction performance” section, an interesting investigation is provided, however authors can improve the estimation power on the sample size by performing evaluation on increasing the sample size in addition to reducing it. For instance, one can simply perform boot strap sampling or more advanced data augmentation methods: https://link.springer.com/article/10.1007/s12065-019-00283-w

Reviewer #2: This is a quite interesting and well-written manuscript, that was a pleasure to read, including a number of useful conclusions and lessons for the researchers who are developing drug response prediction models in cancer cell line panels. This reviewer especially enjoyed the last results section and conclusions at the end of discussion about the required number of cell lines to construct a robust drug response prediction model. However, there remain several issues that will need to be addressed (see below) to make this work more rigid in terms of the statistical analyses, and even more useful for the computational biology community.

Major comments:

1.Although the results are interesting, these are not entirely novel. The authors should put their results and conclusion into the context of previous related works (e.g., PMID: 27444372, PMID: 26274927, PMID: 29016819, PMID: 29186355, PMID: 31208429, PMID: 30704458). They should also reword lines 77-80 in the Introduction and cite the existing studies on this topic. In the discussion, please state whether your results confirm previous observations for tissue-specific contribution to pan-cancer modelling, and what are the novel findings from this work, compared to the previous investigations.

2. Many of the results are based on Spearman rank-correlation, which is a robust measure of rank association between observed and predicted responses. However, since the significance of correlation effects sizes depends on the number of cell lines, correlation values are not directly comparable when comparing pan-cancer and tissue-specific models based on different cell line numbers. It is therefore important to always specify the p-value for the correlations. One can also plot -log (p) to make the comparisons easier to interpret statistically in those plots where the sample sizes are different.

3. The authors need to better justify why AUC (not auROC) and rank correlation were used for binary and continuous prediction problems, respectively. Concordance index (CI) would provide an alternative evaluation metric that can be used in both setups, and would make the comparisons easier to interpret (e.g., Fig. 4C,D). The use of rank correlation in Figure 3 is bit misleading (the same for Figure 4A,B), since Spearman correlation considers variation of ranks, not absolute values or linear fits, like illustrated in these figures. The authors should use Pearson correlation or coefficient of determination in these plots.

4. The section “Sample size advantage…” gives a more direct comparison of pan-cancer and tissue-specific models, where the latter are trained using cell lines from each tissue only. Fig. 5 is a nice comparison, but the authors should specify the number of cell lines used for each tissue-specific models on its x-axis. Also, instead of marking p_1/2, please align p-values to the two columns being compared. The authors should also make one main figure for the comparison of the model features. Suppl. Fig. 3 style is bit hard to access, and simple Venn diagrams might work better for showing overlapping features.

5. The current results focus only on three MEK inhibitors (PD-0325901, GDC-0973 and Selumetinib), out of which only two are shared between the two datasets. The authors are recommended to extend these analyses also other classes of inhibitors to guarantee that the conclusions they make in the end are generalizable also to other drug and target classes. If needed, there are also other large-scale drug testing datasets available, e.g., DGSC and CTRP v.2, which include many dugs and various molecular profiles. The authors could check from PharmacoDB suitable datasets for other kinase inhibitors.

Specific comments:

1. Figure 1C. Statistical testing of the number of sensitive cell lines would make the plot more convincing. Please also justify why threshold of IC50 = 1uM was chosen to determine if a cell line is sensitive to a drug. Ideally, such threshold should depend on a specific drug, relative to other drugs in the cell line.

2. Figure 2D. Comparison of predictions across datasets (red and pink symbols) and within datasets is borderline significant. It would be good to analyze this bit further, as it would be quite surprising result if these prediction accuracies are overall similar, as is currently stated in the results (lines 185-187).

3. Figure 3. The “cigar plots” are very illustrative but they only show the mean observed and predicted log(IC50) from around 0 to 1. However, according to Figure 2C, the observed log(IC50) can be up to 2 and the predicted log(IC50) can be up to 5. Why is there so much difference between the two plots?

4. Figure 6. This is a nice analysis, but may give somewhat simplified view of the pan-cancer model performance as its prediction performance may not only be related to the sample size, but also to the correlations between the tissue groups. This should be further investigated, or at least discussed.

5. Even if the language is good, there are certain bit cryptic sentences and terms that needs to be made more specific; for instance ”group” on lines 92 (“contributions of both group and individual identity”), and line 215 (“this is because the two tissues, at the group level”); please reword and make clearer.

6. Abstract and page 12: “a 22% decrease” is difficult to be understood. What does a 22% decrease in the Spearman correlation coefficient mean? Before and after correlation would be better. Further, since this is an average number, confidence interval or a range of correlations should be reported, too.

7. Abstract “RNA, SNP and CNV data”; these needs to be made more explicit when first time used, e.g., “mRNA expression, point mutations and copy number variation”; the authors should mention in discussion that also other data (e.g. methylation and proteomics) are being used in prediction models.

8. Lines 59-60: For within a cancer type prediction model, the authors should cite such prediction models, e.g., the DREAM7 drug sensitivity prediction challenge in breast cancer cell lines (PMID: 24880487), which is still consider a state-of-the-art in the field of drug response prediction in cancer cell lines.

9. Page 12: the first sentence "five tissues whose within-tissue variability was accurately predicted". Please define what does it mean to be “accurately predicted“? For instance, specify a threshold for the Spearman correlation coefficient to determine whether the prediction is accurate or not.

10. Line 119: Between- and within-tissue performance of pan-cancer MEKi response predictions section. Please emphasize whether these results are based on the pan-cancer model or tissue-specific model (c.f. the next section: Sample size advantage…)? It seems the former, but good to specify in the text.

11. Line 358: ”We further standardized RNA expression data for each gene by linearly scaling values across cell lines to a range between 0 and 1 and shifting the scaled values by subtracting the scaled mean”; the rationale of this post-processing remains unclear and needs to be justified in the Methods section.

12. Methods. Please give more details of the assays of the two studies. For instance, SNP arrays and exome-seq are quite different for detecting point mutations. How does that affect the results? Were the same cut-offs for gene amplifications and deletions used in both studies. No details of the drug assays provided.

13. In the binary problem, the authors use logistic regression with LASSO to first select features. Instead of that, logistic LASSO regression might be a better and more straightforward option which is also implemented in the glmnet package. The authors should consider using that in the revised work.

14. It has been shown that cancer tissue type may directly contribute to the prediction of drug sensitivity in the pan-cancer models. It remained unclear whether the tissue-of-origin was used as predictor in any of the analyses; and if not, the authors need to explain why it was not included in any of the models?

15. Supplementary Table 1 shows boundary lambda in model f_C1 and model f_C2 for the regularised regressions. Please re-tune these parameters. Further, it remains unclear whether the parameters of random forest regression were optimized or not, including "number of tree to grow”, “cutoff”, etc?

Reviewer #3: Inter-tumor heterogeneity in molecular characteristics and phenotypes are well known, but analyzing pan-cancer dataset may yield a uniformly predictive molecular signature among different cancers for certain cancer drugs. Lloyd et al. studied the impact of inter-cancer heterogeneity and intra-cancer heterogeneity on pan-cancer predictions of drug sensitivity. Authors extensively explored the performance of the pan-cancer predictions only for one targeted cancer drug family, MEK Inhibitors. I thus strongly suggest authors to publicly share their analysis codes and datasets used in this study that will enable oncologists and cancer biologists to explore inter-cancer heterogeneity in responses to other anti-cancer therapeutics. in the manuscript, there are paragraphs that need more description and clarification. Please find my comments below.

Comments.

Page 6:

Line 126-129: Figure 1D. For Drug response correlation, a correlation coefficient of PD-901 sensitivities between two cell line panels of 0.81 is lower than correlation coefficients of two different drugs in each of the two studies (r=0.88 between PD-901 and GDC-0973 in Kijin and r=0.83 between PD-901 and Selumetinib). Does the lower correlation coefficient for the same drug imply a larger study-specific batch effect on drug sensitivity measurements?

Page 7:

Line 144-146: Please describe a rationale for a thrould of IC50 <=1nM with which cell lines were stratified as sensitive to a drug.

Line 148: Is logistic regression analysis a regularized logistic regression?

Page 10:

Line 187-189: In Figure 1D, there are many triangles (cross-MEKis prediction) with lower correlation coefficients and auROC values than circles (same MEKi). Although p-value is above 0.05, I don’t think it is reasonable to argue that cross-MEKis performed comparably.

187-190: please specify the ranges of observed spearman correlation coefficients and AUCs in addition to the reported U-test p-values.

Page 12:

Line 204-206: I wonder how many cancer cell lines were originated from those five cancer types in the pan-cancer training dataset. These five cancers may account for the majority of cell lines in the training dataset and show a good prediction performance.

Line 213-216: Please describe How you choose these two cancer types? In Figures 4A and 4B, breast cancer cell lines had truncated IC50 at around 2. It may not make sense to calculate correlation coefficients with this truncated drug sensitivity data.

Line 215-217: To argue this, i think authors need to show that combining two nearest cancer types on PC values and IC50 such as colorectal and stomach does not improve an overall prediction performance.

Line 217: Please clarify what between-tissue signal means. Does it mean difference in gene expression or drug sensitivities between brain and pancreatic cancer cell lines?

Page 15

Line 239-241: It is an important research topic to identify predictive genetic and transcriptomic markers that determines the drug sensitivity of MEKi. It may be a bad idea to elaborate about molecular features that were finally included in each of pan-cancer regression models and how many common features were across the models.

Line 245: This is a very interesting observation. During down sampling of the pan-cancer data, did you randomly sample cell lines or

Page 17:

Line 262-265: I don't think only ovarian cancer cell line will be benefited from a larger pan-cancer data training set. As the sample size of a training set increases, stomach cancer cell lines also seem to show persistently steep increment in tissue-specific prediction performance.

**Have all data underlying the figures and results presented in the manuscript been provided?**

Reviewer #1: Yes

Reviewer #2: Yes

Reviewer #3: Yes

PLOS authors have the option to publish the peer review history of their article (what does this mean?). If published, this will include your full peer review and any attached files.

Reviewer #1: **Yes: **Omid Bazgir

Reviewer #2: No

Reviewer #3: No
---

## [Decision Letter · Decision Letter 1]

12 Nov 2020

Dear Dr. Li,

Thank you very much for submitting your manuscript "Impact of between-tissue differences on pan-cancer predictions of drug sensitivity" for consideration at PLOS Computational Biology. As with all papers reviewed by the journal, your manuscript was reviewed by members of the editorial board and by several independent reviewers. The reviewers appreciated the attention to an important topic. Based on the reviews, we are likely to accept this manuscript for publication, providing that you modify the manuscript according to the review recommendations.

Please address the points of Reviewer 2.

Sincerely,

Ewa Szczurek

Guest Editor

PLOS Computational Biology

Florian Markowetz

Deputy Editor

PLOS Computational Biology

[LINK]

Please address the points of Reviewer 2.

Reviewer's Responses to Questions

**Comments to the Authors:**

Reviewer #1: The authors have addressed all my concerns raised in the first round of review.

Reviewer #2: The authors have done good job in addressing most of my original comments, but there still remain a couple of comments that needs to be addressed better.

Comment 2.1.3. The calculation of AUC for regression task is confusing and questionable. Please report in the figures and text CI instead of AUC, since it works both for regression and classification tasks. Also, do not use the term auROC, rather AUC or AUC-ROC, as those are standards in the field; the use of non-standard terms or calculations of AUC will make it difficult for others to reproduce the results.

Comment 2.2.13: The authors showed similar prediction performances from the logistic regression with LASSO feature selection and logistic LASSO. This might be sufficient for the purpose of drug response prediction only. However, the authors should also make a comparison with respect to feature selection and effect estimation, as these are other important factors for the drug response modelling.

Comment 2.2.14: It remained unclear from the response whether the authors added the 11 binary features for tissue-of-origin as mandatory variables in the regularised regression, i.e., without feature selection for the tissue features. If not, this should be done for proper evaluation of their importance. An additional try is to regress the drug responses on the tissue-label features only, without including any genomic data.

Reviewer #3: I thank the authors for addressing satisfactorily all the reviewer's comments.

**Have all data underlying the figures and results presented in the manuscript been provided?**

Reviewer #1: None

Reviewer #2: Yes

Reviewer #3: Yes

PLOS authors have the option to publish the peer review history of their article (what does this mean?). If published, this will include your full peer review and any attached files.

Reviewer #1: **Yes: **Omid Bazgir

Reviewer #2: No

Reviewer #3: No
---

## [Editor Report · Decision Letter 2]

18 Jan 2021

Dear Dr. Li,

We are pleased to inform you that your manuscript 'Impact of between-tissue differences on pan-cancer predictions of drug sensitivity' has been provisionally accepted for publication in PLOS Computational Biology.

Best regards,

Ewa Szczurek

Guest Editor

PLOS Computational Biology

Florian Markowetz

Deputy Editor

PLOS Computational Biology

---

## [Editor Report · Acceptance letter]

5 Feb 2021

PCOMPBIOL-D-20-00831R2 

Impact of between-tissue differences on pan-cancer predictions of drug sensitivity

Dear Dr Li,

I am pleased to inform you that your manuscript has been formally accepted for publication in PLOS Computational Biology. Your manuscript is now with our production department and you will be notified of the publication date in due course.

With kind regards,

Alice Ellingham
